# Determinants of Health Status and Life Satisfaction among Older South Koreans

**DOI:** 10.3390/healthcare12111124

**Published:** 2024-05-31

**Authors:** Hyun-Chool Lee, Alexandre Repkine

**Affiliations:** 1Political Science Department, Konkuk University, Seoul 05029, Republic of Korea; lhc0609@konkuk.ac.kr; 2Economics Department, Konkuk University, Seoul 05029, Republic of Korea

**Keywords:** healthy and active aging, health status indicators, older South Koreans, life satisfaction, employment opportunities

## Abstract

South Korea is a rapidly aging society with the lowest fertility rates among the OECD economies. It is projected to become a super-aged society in 2025, with the share of individuals older than 65 reaching twenty percent. These developments make it important to analyze the determinants of health outcomes in older individuals. In this study, we identified the determinants of subjective and objective health outcomes among senior individuals in South Korea. We used self-rated health and life satisfaction scores as the two subjective health status indicators, while the number of chronic diseases was the objective one. We ran Tobit multivariate regressions of all three indicators on a set of factors related to the older citizens’ physical, economic, and social characteristics. Active employment status and willingness to work in the future were positively related to self-rated health level but were not statistically related to life satisfaction, while income positively affected both subjective health status indicators. Age did not appear to affect satisfaction with life. Active leisure activities were positively related to both self-rated health and life satisfaction. In contrast, passive leisure, such as watching TV, was negatively related to both health status indicators while being associated with an increased number of chronic diseases. Our findings suggest that older South Koreans view employment primarily as a means of financial support rather than as an opportunity for active social engagement.

## 1. Introduction

South Korea is a rapidly aging society in East Asia. Indeed, as documented in Kim and Kim [1], it took South Korea only fourteen years to become an aged society in 2017 when the share of the population older than sixty-five reached a level of 14%, as compared to twenty-four years in the case of Japan. As discussed by Repkine and Lee [2], the age of sixty-five is considered an old-age threshold in South Korea as it is the basis for designing pension benefits, public transportation discounts, and other assistance offered to senior citizens. Repkine and Lee [2] report fertility rates in South Korea to have dropped to 0.81 in 2021, far below the replacement level of 2.1. In the same year, South Korea’s population registered its first year-on-year decrease of 0.03%, a decline rate that reached a level of 0.06% in 2023. The Korean Statistical Office projects that South Korea will become a super-aged society by 2025, with the share of her citizens older than sixty-five reaching twenty percent [3].

Recognizing the importance of the problem of an aging population in South Korea, studies focusing on older South Koreans have recently increased in number. Thus, among the recent contributions, Lee [4] discussed the general trends of South Koreans’ aging processes, while Lee [5] analyzed their political consequences.

The importance of analyzing the health-related factors that influence the quality of life of the older population is hardly debatable. While the issues of healthy and active aging, as defined by the World Health Organization [6], are generally significant, identifying those factors that influence older people’s health status indicators appears to be especially instructive in a country like South Korea, where the process of population aging has proceeded at an accelerated pace.

First, understanding the determinants of older citizens’ health scores can facilitate efforts to promote healthy aging among older South Koreans, as addressing health risk factors and promoting positive health behaviors can empower older adults to maintain better health and overall well-being as they age. A related discussion is provided by Lee [7] where the author discusses the aging problem in South Korea in an economic context. Second, Jun and Park [8] show that analyzing the determinants of health scores can help identify health disparities within the older South Korean population. Certain demographic, socio-economic, and health-related factors may contribute to these disparities, highlighting the need for targeted intervention. Third, a comprehensive empirical analysis by Brenowitz et al. [9] suggests that health score analysis also helps to predict long-term health outcomes, including mortality and functional decline among older adults. Finally, Wang et al. [10] suggest that by identifying determinants associated with self-reported or objective health scores, policymakers can prioritize interventions aimed at improving health outcomes and quality of life among older adults.

In this study, we relate older South Koreans’ self-rated health scores, life satisfaction indices, and the number of chronic diseases to several socio-economic and lifestyle characteristics, such as physical activity levels, diet, smoking habits, alcohol consumption, social relationships, labor market involvement, and psychological well-being.

Our choice of self-rated health scores as one of the three dependent variables in our analysis was conditioned by the fact that existing research has demonstrated a strong association between self-rated health scores and healthy aging outcomes. Thus, Jylhä [11] demonstrates that older adults who report higher self-rated health scores tend to exhibit a better functional status and higher levels of social engagement. Benyamini et al. [12] argue that individuals with positive perceptions of their health are more likely to regularly exercise and maintain healthy diets. At the same time, as demonstrated in Idler and Kasl [13], older individuals who rate their health as fair or poor are more likely to experience disability and mortality.

In addition to self-rated health scores and the number of chronic diseases that may be considered subjective and objective health measures, respectively, we employ another subjective measure of well-being, the life satisfaction score, as it has also been shown to be positively associated with the quality of aging. For instance, one meta-analysis conducted by Pinquart and Sörensen [14] explored the relationship between life satisfaction and successful aging among older adults and found that higher levels of life satisfaction were significantly correlated with better physical health and greater social engagement, all of which are indicative of a higher quality of aging. Similarly, Okun et al. [15] highlighted the importance of life satisfaction to the general well-being levels among older adults.

We believe our analysis fills several gaps in the existing literature. First, to our knowledge, ours is the first study to look at the determinants of subjective and objective health outcomes in older South Koreans in a large sample covering more than ten thousand individuals. Second, the determinants of health outcomes we are employing constitute a comprehensive group, thus acknowledging the multidimensional nature of the aging process, which helps us analyze the impact of select determinants on health outcomes in an isolated fashion. Thus, a review of the surveys by Bize et al. [16] and Hamer and Stamatakis [17], based on a sub-sample of 921 participants in the Health Survey of England conducted in 2008, document a positive association between regular physical activity and higher self-rated health scores while not accounting for the respondents’ marital or employment status or the effect of physical activity on the perceived life satisfaction. Similarly, Agborsangaya et al. [18] found evidence of a strong correlation between the number of chronic diseases and poor lifestyle habits, such as sedentary behavior, smoking, and excessive alcohol consumption, while omitting the extent of involvement of the older individuals in social or economic life. Third, our analysis attempts to bridge the gap between medical and social sciences by explicitly acknowledging the importance of both physical and socio-economic characteristics to the health outcomes of older individuals.

This study is organized as follows. In Section 2, we describe the survey employed as a basis for our empirical work, the variables employed in this study, and the statistical methods involved. We present our empirical results in Section 3 and discuss them in Section 4. Section 5 summarizes and concludes.

## 2. Materials and Methods

### 2.1. Survey Data Description

To obtain our empirical results, we analyzed responses from the National Survey of Older Koreans ([19]) conducted by the Korean Institute for Health and Social Affairs (KIHASA) in 2020 and covering 10,097 South Korean individuals aged sixty-five and older. The survey received approval from the KIHASA’s Institutional Review Board to ensure compliance with the ethical norms. The approval, No. 117071, was received on the 6 August 2020.

The pilot survey was conducted by KIHASA researchers and three trained surveyors from 26 June 2020 to 12 July 2020, involving 67 participants. Conducted amid the COVID-19 pandemic, the survey methodology was reviewed in advance to ensure safety and reliability. 

This survey was conducted every three years starting in 2008. At the age of sixty-five, South Korean employees are entitled to receive their pension payments, public transportation discounts, and the like, which makes it a natural threshold for defining older age. Interestingly, historically, the age of sixty was considered the start of old age in South Korea.

A team of twenty-eight experts specializing in various fields collaborated to design the survey’s questions. Subsequently, the questions underwent scrutiny by the KIHASA’s Institutional Review Board before receiving official approval. Over a hundred interviewers visited households where at least one member had reached the age of sixty-five, conducting tablet-assisted interviews either with the older individuals themselves or their representatives. Following the interviews, each answer sheet underwent screening to address inconsistencies and missing responses, with corrections made if necessary. SPSS software (version 25) helped identify outliers, non-sampling errors, and invalid entries. While obvious errors were rectified, entries deemed dubious were classified as missing values.

To ensure that the survey sample accurately reflected the demographic composition of South Korea’s older population, the research team referred to the Korean Population and Housing Census conducted by the Korean National Statistical Office (NSO). This comprehensive census provides detailed demographic data, including population numbers categorized by age, gender, and other characteristics across seventeen South Korean city and provincial regions.

The KIHASA employed a stratified random sampling approach to construct the survey dataset. The population was first divided into strata based on data from the Korean Population and Housing Census conducted by the Korean National Statistical Office (NSO). Within each stratum, participants were randomly selected, ensuring that the sample was representative of the population’s structure. Sample sizes for each stratum were determined to proportionately represent the population within each of the seventeen regions, resulting in a sample of 10,097 respondents. Region-specific sample sizes were allocated using the adjusted square-root two-stage allocation method, as outlined by Schafer [20].

Our empirical analysis is based on a subset of the sample, consisting of 9306 individuals, that resulted from excluding observations with missing responses to one or more questions of interest. We applied Little’s [21] test to assess whether the missing data were missing completely at random (MCAR), ensuring independence from the observed data. While the test statistic of 1309 suggested that the missing data were not MCAR, given that the removed individuals comprised less than eight percent of the original sample, our statistical analysis was unlikely to be significantly biased, as discussed by Raymonds and Roberts [22].

### 2.2. Variables and Measurements

We employed the self-rated health score, life satisfaction index, and the number of chronic diseases in older adults as dependent variables representing the state of respondents’ physical and psychological health. We describe these measures and define their usage in Section 2.2.1. In Section 2.2.2 to Section 2.2.6 we describe the five groups of health status indicators’ determinants, or independent variables, that we believe are related to the three health measures. We describe our statistical analysis in Section 2.2.7.

#### 2.2.1. Health Outcomes

As argued by Voukelatou et al. [23], health is one of the six dimensions of human well-being, together with the quality of job opportunities, socioeconomic development, environment, safety, and political climate. While objective health status indicators, such as the number of chronic diseases, are related to good health, subjective health status indicators are correlates of good health, too. Thus, the self-rated health (SRH) indicator, measured on a Likert scale from 1 (poor) to 5 (excellent), was employed by Meng et al. [24] and French et al. [25] and was shown to be related to objective health measurements and the general state of body and mind. Life satisfaction measured on a Likert scale was shown to be related to the general state of health by Grant and Steptoe [26].

For the objective health measurement, we used binary answers to seventeen questions on the presence of chronic diseases such as high blood pressure or diabetes. The overall objective health status indicator is the sum of the individual scores on each one of the seventeen disease-related yes-or-no questions. The number of chronic diseases is widely known to be closely related to health outcomes. A study by Tinetti et al. [27] provides a comprehensive discussion of this relationship on the basis of the general population, while Pearson et al. [28] present an empirical investigation in the context of aging.

#### 2.2.2. Physical Characteristics

The respondents’ physical characteristics were age, sex, and body mass index (BMI), defined as a ratio of one’s weight in kilograms to the square of one’s height in meters. BMI was shown to be related to the measures of both subjective and objective health, as demonstrated by Stienen et al. [29] and Chang et al. [30]. We included the age variable as the number of chronic diseases is expected to grow with age. Studies, such as that by Joshanloo and Jovanovic [31], have found that men and women score differently in terms of their subjective life satisfaction, which is why we included a gender binary variable into the set of the health status indicators’ determinants.

#### 2.2.3. Socio-Economic Variables

Labor market and income-related characteristics were captured by five variables: employment status, willingness to work in the future, total number of years worked, total income, and educational achievement. Having a paid job was shown to be related to both subjective and objective health outcomes of older people in a study by Axelrad et al. [32], who argued that unemployed senior citizens often feel they are neglected by society. An argument saying that this sense of neglect is exacerbated by a drop in income is suggested by Bonsang and Klein [33]. A study by Min and Cho [34] provides interesting insights into the South Korean working culture, emphasizing the fact that a significant number of older South Koreans value the idea of staying active and contributing to society through work even after retirement age. We, therefore, believe that older individuals’ characteristics related to their work, education, and income are crucial determinants of their physical health and life satisfaction.

The “total number of years worked” and “willingness to work in the future” variables captured the time dimension of an older person’s employment characteristics. Total income was measured as the general living standard. In several studies, such as that by Wagg et al. [35], educational achievement was found to be positively related to healthy aging. Kim and Park [36] provide a general discussion of the relationship between education and health outcomes in older South Koreans, and Park and Lee [37] look at an association between older South Korean adults’ educational background and self-reported health status.

#### 2.2.4. Relationships with Friends and Family

The strength and quality of family ties were shown to be related to health outcomes in studies by Umberson and Thomeer [38] and Mehrabi and Beland [39], motivating us to employ the spouse’s health measured on a Likert scale. This included two binary variables assuming the value of one in case the respondent had a conflict with his or her spouse or offspring during the past year and the frequency of visits with friends. Including these two variables in the set of determinants of older South Koreans’ health outcomes is particularly significant in the context of Korean culture that emphasizes familial and social ties. A study by Park et al. [40] presents a comprehensive discussion of the importance of friends and family ties to the health outcomes among older South Korean individuals.

#### 2.2.5. Lifestyle Characteristics

Studies by Sowa et al. [41] and Abud et al. [42] provide empirical evidence for the importance of lifestyle characteristics to the health outcomes of the aged population. Bize et al. [16] and Hamer and Stamatakis [17] demonstrated a significant role played by regular physical exercise in improving self-perceived health and overall well-being. Studies, such as that by Kaplan et al. [43], have also consistently shown that older adults who smoke or consume alcohol excessively are more likely to report lower self-rated health compared to non-smokers and moderate drinkers. In the Korean context, Yoon et al. [44] provide insights into the importance of leisure activities for promoting health and well-being in older adults, emphasizing the potential benefits of engaging in diverse leisure activities for maintaining physical, cognitive, and emotional health.

We represented lifestyle’s physical dimension with a binary variable equal to unity in case the respondent exercises regularly, a binary variable representing smokers, and the frequency of consuming alcoholic beverages. Pastime characteristics were represented by a binary variable equal to unity in the case when the respondent traveled somewhere or engaged in general leisure activities during the past year and the daily number of hours spent watching TV. Finally, two binary variables identified the owners of a smartphone and a computer.

#### 2.2.6. Social Interactions

The importance of social interactions to older individuals’ physical and mental health was confirmed by numerous studies, such as that by Asante and Karikari [45]. The association between participation in leisure activities, including hobbies and social activities, and the risk of dementia among older adults was examined by Verghese et al. [46]. The importance of social activities to physical health was corroborated by the study of Douglas et al. [47].

Given the data availability of the survey at our disposal, we employed one binary variable to capture the quality of social interactions that were equal to unity in the case when a respondent was part of an informal hobby group for the past one year.

#### 2.2.7. Statistical Analysis

Since the three variables measuring health outcomes are censored both from below and above, we ran a multivariate Tobit regression to evaluate the contribution of the determinants discussed above to the health outcomes.

Denote *Y* as the value of one of our three health status indicators. In the case of self-rated health and life satisfaction $Y\in \left[1,5\right]$ while $Y\in \left[1,14\right]$ in the case of chronic diseases. Rather obviously, the higher values of *Y* indicate better health outcomes, while they are associated with worse health outcomes in the case of the number of chronic diseases. Let $\vec{X^{\prime }}$ be a vector of health status indicators’ determinants and $\vec{\beta }$ a vector of the corresponding coefficients. While it is possible in principle to estimate the elements of $\vec{\beta }$ by running an OLS regression $Y_{i}=\vec{X^{\prime }}\vec{\beta }+\varepsilon_{i}$ where $\varepsilon_{i}$ are identically and independently distributed random shocks, the resulting predicted values of the dependent variable $Y_{i}$ will not necessarily belong to the specified range of $Y_{i}$, which is why an OLS estimation is likely to produce biased and inefficient coefficient estimates, as shown by Amemiya [48].

A standard way of dealing with this problem is to consider a latent variable $Y_{i}^{*}$ that is linked to the observed variable $Y_{i}$ as follows:(1)$$Y_{i}=\left\{\begin{array}{l} 0,Y_{i}^{*}\leq a\\ Y_{i}^{*},Y_{i}^{*}\in \left[a,b\right]\\ b,Y_{i}^{*}\geq b \end{array}\right.$$

In this case, the regression specification $Y_{i}=\vec{X^{\prime }}\vec{\beta }+\varepsilon_{i}$ becomes
(2)$$Y_{i}^{*}=\vec{X^{\prime }}\vec{\beta }+u_{i}$$
where $u_{i}\neq \varepsilon_{i}$ are random i.i.d. shocks.

Model (2) is described by Wooldridge [49] and is known as a two-limit Tobit model. We used STATA software, version 13.1, to estimate the values of $\vec{\beta }$ in (2).

## 3. Results

Table 1 presents the survey respondents’ objective and subjective health characteristics, namely, their self-rated health, life satisfaction level, and the number of chronic diseases. Half the respondents reported satisfactory health levels, were satisfied with life, and suffered from two or more chronic diseases.

Table 2, Table 3 and Table 4 summarize the respondents’ physical, socio-economic, family, and lifestyle characteristics. Out of the ten thousand participants, 60.04% were women. The respondents’ age varied from 65 to 102, with a median value of 72. About half the respondents had a BMI of 23.4 or below, which was well within the healthy range.

Table 3 shows the respondents’ physical characteristics. The median income of our survey participants was KRW 8.9 million, an equivalent of US 6542. One-third of the survey participants had a paid job, while one-quarter of them expressed willingness to work in the future. Elementary school was the highest educational achievement for half the respondents in our sample.

Table 4 presents the family-related characteristics of the survey respondents. Most of the participants were married, and only 15% lived together with their children. The overwhelming majority reported having had no recent conflicts with either their spouse or children.

Table 5 shows the respondents’ lifestyle characteristics. The majority were non-smokers and drank alcohol only occasionally. More than half the respondents reported regular exercise, while more than three-quarters engaged in some sort of leisure activities, watching TV programs for an average of four hours a day.

Table 6, Table 7, Table 8 and Table 9 present the results of our multivariate Tobit regression analysis. Table 6 deals with the respondents’ physical characteristics. BMI, or its square, was estimated to be statistically significant in the case of all three health measures, while age was not relevant for life satisfaction. Interestingly, gender was estimated to be statistically relevant only for the number of chronic diseases but not for the subjective health measures.

Table 7 shows regression results for five socio-economic characteristics. A higher income was uniformly associated with better health outcomes. Better education appeared to be positively associated with subjective health measures while being irrelevant for the number of chronic diseases. Having a paid job apparently increased self-rated health, while the working experience was not found to be statistically relevant for any health measure.

Table 8 reports the regression results for the “Friends and Family” group of determinants. Conflicts with either offspring or spouse, the spouse’s health, and visiting friends were estimated to be statistically significant in the case of all three measures with the expected signs.

Table 9 presents the Tobit regression results for the respondents’ lifestyle characteristics. Owning a smartphone or a computer did not appear to be relevant for the respondents’ health. Surprisingly, neither was smoking. While expectedly negatively associated with one’s life satisfaction, the frequency of drinking alcohol was negatively correlated with the number of chronic diseases. Longer hours of watching TV were statistically significant for all three health measures, suggesting a negative association. Traveling, enjoying leisure, and being part of a hobby group were either positively associated with better health outcomes or were irrelevant to them.

## 4. Discussion

The purpose of this study was to identify factors that determine the subjective and objective health status indicators of older South Koreans based on a comprehensive survey of ten thousand respondents. While statistically significant factors affecting either self-rated health or life satisfaction had expected signs, the sets of these determinants appeared to be different between the two subjective health measures. This finding was further corroborated by the fact that Pearson’s correlation coefficient between self-rated health and life satisfaction indices in our sample was rather low at 40%.

We find it rather alarming that life satisfaction appeared to be unrelated to one’s employment status or willingness to work in the future while being positively related to the level of income. These research results are in contrast to Lim and Cho [50] who found that economic activity has a positive effect on the quality of life of the elderly. A study by Jung [51] also found that middle-aged and older workers aged fifty or older had high levels of depression when their education level did not match the status and quality of their jobs. As a result of this perception, their self-rated health status indicators were low as well. The apparent lack of statistical association between older South Koreans’ involvement in the labor market and their life satisfaction is rather alarming in a country where, as argued by Min and Cho [34], older South Koreans attach a significant emphasis to remaining active and to contributing to society through work. We are thus inclined to believe that an older South Korean person having a job at an older age is likely considered to be more of a means to make ends meet rather than a way to continue being an active member of society.

Indeed, the areas in which people aged 65 or older worked the longest were manual labor (48.7%), agricultural and fishery (13.5%), services (12.2%), management (8.8%), maintenance (5.6%), and sales (4.7%). As argued by Ji [52], the fact that most older South Koreans engage in routine labor jobs limits their opportunities to utilize the knowledge and skills accumulated during their lifetime, leading to low self-esteem, increased frustration, and low life satisfaction.

If true, our findings are rather disappointing given the number of programs recently initiated by the South Korean government in order to increase older individuals’ participation in social life, such as lifelong educational programs. In our view, one clear policy implication would be to focus on providing older South Koreans with meaningful job opportunities that they would not merely consider as a source of income, but as a means of social involvement as well.

We find it rather interesting that while having a paid job and the number of chronic diseases were not found to be statistically associated, willingness to work in the future was negatively correlated with the number of chronic diseases, which is likely indicative of a causal link between a positive attitude towards possible employment and the physical health outcomes. This causality, however, cannot be properly established within the cross-sectional framework of this study, so we leave it for our future research.

The “Friends and Family” group was the only one whose constituent determinants were estimated to be statistically significant in determining all three measures of health. Thus, conflicts with offspring and spouses are detrimental to both self-rated health and life satisfaction while being positively associated with the number of chronic diseases. These results are consistent with the findings of an existing study showing that the effect of spousal support has the greatest effect on the subjective health status of the elderly, based on an analysis of eighty papers in South Korean academic journals, as documented by Rhee [53]. Similarly, a healthy spouse is positively associated with self-rated health and life satisfaction while being negatively associated with the number of chronic diseases. These findings underscore the importance of maintaining traditional family ties and values. The government’s resolve to uphold these values would contribute to a healthy aging society.

While our empirical results accord reasonably well with the findings reported by the existing literature, there are notable differences. Thus, similarly to Marengoni et al. [54], we found a more advanced age to be positively correlated with the number of chronic diseases, while its association with self-rated health was negative, a finding reported by Harris et al. [55]. However, our finding of a positive association between BMI and subjective health outcomes contrasts with the findings of Zajacova and Dowd [56], who reported a negative correlation. Our explanation of the discrepancy is that 75% of the respondents in our sample had a BMI well within the healthy range, i.e., BMI < 25, with increased BMI having to do more with a sufficient and healthy diet rather than problems with obesity.

Expectedly, a number of studies, such as that by Wagg et al. [35], found a positive association between the level of income and healthy aging, as higher incomes are conducive to higher quality health care services and better leisure opportunities. Our finding of a positive correlation between income and the number of chronic diseases is rather disconcerting and needs to be explained. The most likely explanation seems to be the survivorship bias, i.e., a situation when higher-income individuals survive longer because they have access to better healthcare while at the same time developing more chronic diseases with age.

In contrast to Axelrad et al. [32], who found a positive influence of employment at an older age on subjective health outcomes, we did not find a statistically significant association between older individuals’ employment status and their life satisfaction; although, similarly to Axelrad et al. [32], we found a positive association with self-rated health. Given the high importance attributed by the older South Koreans to working at an old age reported by Min and Cho [34], the apparent lack of an association between one’s employment status and life satisfaction is indicative of the fact that older South Koreans consider having a job as a means to make ends meet rather than a means to remaining active members of the society.

Finally, our finding of a negative association between the frequency of consuming alcohol and the number of chronic diseases is in stark contrast with most of the existing studies examining this association, such as that by Shield et al. [57], which emphasizes the detrimental effects of increased alcohol consumption on the incidence of cancer, cardiovascular, or liver diseases. One explanation may be in terms of the healthy user bias that happens when individuals who drink alcohol moderately engage in health-promoting activities such as better diets and regular exercise. This explanation seems to be plausible in light of the fact that 77% of the respondents reported drinking alcohol less than three times a week.

Our findings are meaningful in light of the recent growth of interest in promoting active and healthy aging among both researchers and government policymakers. We found that an active pastime such as traveling was positively associated with life satisfaction and self-rated health, while a passive leisure activity such as watching TV was inversely related to the two health measures and was positively related to the number of chronic diseases; therefore, we believe government policies aimed at promoting active leisure among senior citizens are critical, especially given the fast pace of aging in the South Korean society.

Our study has certain limitations. For instance, both the subjective and objective health measures employed in our study are reported by the respondents themselves, which may render these measures biased in the case of the respondents suffering from neurocognitive disorders, which may hamper or distort their memory. Second, because the survey was conducted in the year 2020 during the COVID-19 pandemic and required explicit consent on the part of the respondents, some participants with pre-existing medical conditions or concerns about their health were likely to refuse to take the survey, thus creating a sample selection bias. Third, the COVID-19 epidemic itself could have played a role by introducing a bias in those participants who had been affected by the virus prior to responding to the survey questions. 

Certain of our empirical findings, there appears to be a need for further investigation. Thus, it appears worthwhile examining the survivorship bias hypothesis in the context of a positive association we found between income levels and the number of chronic diseases. Secondly, we plan to test the validity of our preliminary conclusion that older South Koreans view employment opportunities as a means to survive rather than a way to remain active members of society. Finally, the healthy user bias hypothesis seems interesting to analyze in light of our finding of a negative association between alcohol consumption and the number of chronic diseases.

## 5. Conclusions

In this study, we related three measures of older individuals’ health to a comprehensive set of determinants consisting of five groups. The determinants affecting self-rated health, life satisfaction, and the number of chronic diseases were estimated to be different, although some characteristics, such as conflicts with spouses and offspring or spouse’s health, were estimated to be statistically significant for all three health status indicators in our study. 

One of the key empirical results we obtained is that having a paid job for an older South Korean person was more likely to be a means of making ends meet rather than a way to feel more satisfied with life. We, therefore, conclude that policy efforts are needed to move away from the paradigm that focuses on the provision of jobs per se to the elderly and towards an approach that is centered on providing quality jobs, i.e., the ones that fit older workers’ careers. In South Korea, most elderly people appear to work chiefly in order to earn living expenses. Yet, half of the elderly are poor, making it essential to provide them with more job opportunities. However, this is unlikely to help reach the goal of improving the quality of life of the elderly through work. Instead, the focus should be on expanding the quality of jobs that fit the older workers’ career experience and knowledge. Similar to the conclusion drawn by Ji [52] and Chung et al. [58], we believe it is desirable to promote policies that support the provision of jobs that allow older people to work as proud members of society and realize their selves rather than just trying to meet their basic economic needs.

A positive work attitude is nevertheless found to be statistically associated with higher levels of self-rated health and a lower number of chronic diseases, suggesting that even low-quality jobs may be pivotal contributors to the older South Koreans’ health outcomes.

Our findings imply that the government policies aimed at strengthening traditional family values, providing meaningful job opportunities for older citizens, and enhancing their travel and active leisure opportunities will go a long way in increasing the quality of life of South Korea’s senior individuals as the labor market conditions appear to be about as significant to their health outcomes as are the older South Koreans’ physical and lifestyle characteristics.

Finally, it appears critical for the South Korean government to enforce regular health checks for the older population to provide updates on the determinants of health status, which may serve as the basis for government policy. Operationalizing these health screening programs is beyond the scope of this study, being a subject of separate research.

## Figures and Tables

**Table 1 healthcare-12-01124-t001:** Summary Statistics of the Dependent Variables.

Variable	Variable Type	% Observations	Min	Max	Median	SD
Health Outcomes						
Self-Rated Health	Likert scale: 1 (very poor) … 5 (very good)	1: 201 (2.03%) 2: 1659 (16.72%) 3: 3120 (31.45%) 4: 4507 (45.43%) 5: 433 (4.36%)	1	5	3	0.875
Number of Chronic Diseases	Integer		0	16	2	1.450
Life Satisfaction	Likert scale: 1 (very poor) … 5 (very good)	1: 45 (0.45%) 2: 666 (6.71%) 3: 4069 (41.02%) 4: 4722 (47.60%) 5: 418 (4.21%)	1	5	4	0.704

**Table 2 healthcare-12-01124-t002:** Summary Statistics of the Physical Characteristics.

Variable	Variable Type	% Observations	Min	Max	Median	SD
Physical Characteristics						
Age	Integer		65	102	72	6.632
Sex	Binary 0: man 1: woman	0: 4035 (39.96%) 1: 6062 (60.04%)	0	1		
Body Mass Index	Real		17.631	30.487	23.495	2.615

**Table 3 healthcare-12-01124-t003:** Summary Statistics of the Socio-Economic Characteristics.

Variable	Variable Type	% Observations	Min	Max	Median	SD
Socio-Economic Characteristics						
Employment Status	Binary 0: does not have a paid job 1: has a paid job	0: 6312 (62.51%) 1: 3785 (37.49%)	0	1		
Willingness to Work in the Future	Binary 0: does not want to work 1: wants to work	0: 6189 (61.07%) 1: 3945 (38.93%)	0	1		
Number of Years Worked	Integer		0	75	20	15.73
Total Income	mn KRW/year		0	598	8.89	23.263
Education Level	Integer 1 (no school) … 4 (elementary school) … 7 (graduate degree)	1: 341 (3.38%) 2: 830 (8.22%) 3: 3377 (33.45%) 4: 2369 (23.46%) 5: 2668 (26.42%) 6: 203 (2.01%) 7: 309 (3.06%)	1	7	4	1.243

**Table 4 healthcare-12-01124-t004:** Summary Statistics of the Family Characteristics.

Variable	Variable Type	% Observations	Min	Max	Median	SD
Friends and Family						
Spouse’s Health	Likert scale: 1 (very poor) … 5 (very good)	1: 122 (2.06%) 2: 683 (11.52%) 3: 1650 (27.82%) 4: 3128 (52.75%) 5: 347 (5.85%)	1	5	3	0.849
Conflict with Offspring	Binary 0: no conflict 1: was conflict	0: 9584 (94.92%) 1: 513 (5.08%)	0	1		
Conflict with Spouse	Binary 0: no conflict 1: was conflict	0: 5036 (86.11%) 1: 812 (13.89%)	0	1		
Visiting Friends	Integer 1 (rarely) … 7 (frequently)	1: 486 (4.81%) 2: 290 (2.87%) 3: 668 (6.62%) 4: 1484 (14.70%) 5: 1702 (16.86%) 6: 2826 (27.99%) 7: 2641 (26.16%)	0	6	5	1.645

**Table 5 healthcare-12-01124-t005:** Summary Statistics of the Lifestyle Characteristics.

Lifestyle Characteristics						
Traveling	Binary 0: no traveling 1: travels	0: 7555 (74.82%) 1: 2542 (25.18%)	0	1		
Daily Hours Watching TV	Integer		1	18	4	1.947
Leisure	Binary 0: no leisure activities 1: has leisure activities	0: 2158 (21.37%) 1: 7939 (78.63%)	0	1		
Has Smartphone	Binary 0: no phone 1: has phone	0: 4621 (45.77%) 1: 5476 (54.23%)	0	1		
Has Computer	Binary 0: no computer 1: has computer	0: 9034 (89.47%) 1: 1063 (10.53%)	0	1		
Smoker	Binary 0: not a smoker 1: smokes	0: 8993 (89.07%) 1: 1104 (10.93%)	0	1		
Drinking Frequency	Integer 1: almost never 8: daily	1: 6399 (63.38%) 2: 529 (5.24%) 3: 872 (8.64%) 4: 894 (8.85%) 5: 752 (7.45%) 6: 455 (4.51%) 7: 85 (0.84%) 8: 111 (1.10%)	1	8	1	1.747
Regular Exercise	Binary 0: no 1: yes	0: 4855 (48.08%) 1: 5242 (51.92%)	0	1		
Hobby Groups	Binary 0: not member 1: is member	0: 5940 (58.83%) 1: 4157 (41.17%)	0	1		

**Table 6 healthcare-12-01124-t006:** Tobit Regressions: Physical Characteristics.

Variable	Self-Rated Health	Life Satisfaction	Number of Chronic Diseases
Physical Characteristics			
Age	−0.010 (0.002) *** [−0.014, −0.005]	−0.003 (0.002) [−0.007, 0.001]	0.028 (0.005) *** [0.018, 0.037]
Sex	0.014 (0.027) [−0.038, 0.066]	0.018 (0.022) [−0.026, 0.062]	0.313 (0.056) *** [0.203, 0.423]
Body Mass Index (BMI)	0.159 (0.037) *** [0.087, 0.232]	0.078 (0.031) ** [0.017, 0.139]	−0.103 (0.078) [−0.255, 0.049]
BMI^2^	−0.003 (0.0008) *** [−0.005, −0.002]	−0.002 (0.0006) *** [−0.003, −0.0004]	0.004 (0.002) ** [0.001, 0.007]

Note: standard errors in round parentheses. *** stands for a 1% significance level, ** for 5%. 95% confidence intervals are in square brackets.

**Table 7 healthcare-12-01124-t007:** Tobit Regressions: Socio-Economic Characteristics.

Variable	Self-Rated Health	Life Satisfaction	Number of Chronic Diseases
Socio-Economic Characteristics			
Employment Status	0.132 (0.028) *** [0.078, 0.186]	0.005 (0.023) [−0.040, 0.050]	−0.037 (0.058) [−0.151, 0.077]
Willingness to Work in the Future	0.091 (0.027) *** [0.039, 0.144]	−0.028 (0.022) [−0.071, 0.016]	−0.125 (0.057) ** [−0.236, −0.014]
Number of Years Worked	−0.0005 (0.0008) [−0.002, 0.001]	−0.00008 (0.0006) [−0.001, 0.001]	0.0003 (0.002) [−0.003, 0.004]
Total Income	0.020 (0.005) *** [0.010, 0.031]	0.029 (0.004) *** [0.020, 0.037]	0.021 (0.010) ** [0.001, 0.041]
Education Level	0.071 (0.010) *** [0.051, 0.092]	0.067 (0.009) *** [0.050, 0.084]	−0.022 (0.022) [−0.065, 0.020]

Note: standard errors in round parentheses. *** stands for a 1% significance level, ** for 5%. 95% confidence intervals are in square brackets.

**Table 8 healthcare-12-01124-t008:** Tobit Regressions: Family Characteristics.

Variable	Self-Rated Health	Life Satisfaction	Number of Chronic Diseases
Friends and Family			
Spouse’s Health	0.401 (0.013) *** [0.375, 0.426]	0.223 (0.011) *** [0.202, 0.244]	−0.351 (0.027) *** [−0.404, −0.298]
Conflict with Offspring	−0.097 (0.051) * [−0.197, 0.002]	−0.100 (0.042) ** [−0.183, −0.017]	0.210 (0.107) ** [0.001, 0.420]
Conflict with Spouse	−0.107 (0.031) *** [−0.168, −0.046]	−0.274 (0.026) *** [−0.325, −0.223]	0.274 (0.066) *** [0.145, 0.403]
Visiting Friends	0.022 (0.007) *** [0.009, 0.035]	0.043 (0.005) *** [0.032, 0.054]	−0.077 (0.014) *** [−0.104, −0.050]

Note: standard errors in round parentheses. *** stands for a 1% significance level, ** for 5%, * for 10%. 95% confidence intervals are in square brackets.

**Table 9 healthcare-12-01124-t009:** Tobit Regressions: Lifestyle Characteristics.

Variable	Self-Rated Health	Life Satisfaction	Number of Chronic Diseases
Lifestyle Characteristics			
Traveling	0.077 (0.024) *** [0.029, 0.124]	0.035 (0.020) * [−0.005, 0.075]	0.002 (0.051) [−0.098, 0.103]
Daily Hours Watching TV	−0.019 (0.006) *** [−0.030, −0.007]	−0.016 (0.005) *** [−0.026, −0.006]	0.060 (0.012) *** [0.036, 0.084]
Leisure	0.020 (0.027) [−0.033, 0.074]	0.098 (0.023) *** [0.054, 0.143]	0.054 (0.058) [−0.059, 0.167]
Has Smartphone	−0.007 (0.025) [−0.057, 0.043]	0.032 (0.021) [−0.010, 0.074]	−0.060 (0.054) [−0.165, 0.045]
Has Computer	−0.033 (0.032) [−0.095, 0.030]	0.020 (0.027) [−0.032, 0.072]	0.004 (0.067) [−0.129, 0.136]
Smoker	0.027 (0.032) [−0.036, 0.090]	−0.035 (0.027) [−0.087, 0.018]	0.029 (0.068) [−0.105, 0.162]
Drinking Frequency	0.010 (0.006) [−0.003, 0.022]	−0.011 (0.005) ** [−0.022, −0.001]	−0.024 (0.014) * [−0.051, 0.002]
Regular Exercise	0.057 (0.021) *** [0.015, 0.099]	−0.002 (0.018) [−0.037, 0.033]	0.007 (0.045) [−0.081, 0.095]
Hobby Groups	0.111 (0.024) *** [0.063, 0.158]	0.072 (0.020) *** [0.033, 0.112]	0.101 (0.051) [0.001, 0.201]
Constant Term	0.780 (0.496) [−0.192, 1.753]	1.828 (0.416) *** [1.012, 2.644]	0.550 (1.043) [−1.494, 2.594]

Note: standard errors in round parentheses. *** stands for a 1% significance level, ** for 5%, * for 10%. 95% confidence intervals are in square brackets.

## Data Availability

The dataset for this study is available from the authors immediately upon request.

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
