# Peer review of "Determinants of Health Status and Life Satisfaction among Older South Koreans"

_healthcare, 2024, doi:10.3390/healthcare12111124_

Round 1
Reviewer 1 Report
Comments and Suggestions for Authors
I would like to thank you again for the honor of being involved in the review of the article entitled "Determinants of Health Status and Life Satisfaction Among Older Koreans", which aimed to identify the determinants of subjective and objective health outcomes among older people in South Korea.
I would like to congratulate the authors on this article, whose contribution may lead to a better understanding of ageing and the determinants of health status and life satisfaction among older people.
The ageing of the world's population, and especially the rapid ageing of the population in South Korea, will pose multidimensional challenges that require strategic planning and appropriate policies. These should ensure healthy, active, and dignified ageing for all older people. This demographic ageing will pose several challenges to Korean society, such as pressure on health and care systems, labor shortages, strain on pension and social security systems, increased social isolation, a possible increase in the number of older people who are ill or suffering from diseases, and an increase in health-related expenditure. The study could therefore lead to a better understanding of the needs of older people, improve public health policies, optimize health care, reduce inequalities, and promote healthy and active ageing.
Our comments will focus on both the form and content of the manuscript to highlight the improvements that need to be made.
Title: This does not indicate the type of study. Would it be possible for the authors to add this information to better guide the reader?
I. Introduction
We found that the introduction was particularly long and started in a rather odd way. We suggest that the authors restructure their introduction by starting with the context rather than the aim. The way in which others are quoted does not seem appropriate to us. You must go to the references each time to find the author(s) mentioned, which makes reading tedious. Examples: lines 32, 34, 38, 48, 49, 55, 60, 67, 69, 72, 78, etc....
II. Materials and Methods
This section seems to us to be well described.
III. Results
The presentation of the results seems to duplicate the tables. The authors should limit themselves to the socio-demographic characteristics of the study population and the determinants, in a short text that does not duplicate the data in the tables.
IV. Discussion
Limitations should be included in this section. The subjective and objective criteria chosen are all related to the memory of older adults, which could unfortunately lead to bias if these people suffer from neurocognitive disorders.
Author Response
We wish to thank Reviewer 1 for his or her very helpful comments.
Please find our detailed response in the attached file.
Thank you,
Hyun-Chool Lee, Alexandre Repkine and Luwen Zhang

Reviewer 2 Report
Comments and Suggestions for Authors
I have reviewed the manuscript entitled “Determinants of Health Status and Life Satisfaction Among Older Koreans”. As I have seen it is addressed as a crucial topic, However, many comments must be considered to improve the quality of the study. I hope the authors don’t feel offended by my comments which only were raised to help them.
· The abstract is good but better to start with the background of the study before the aim of this study.
· The keywords missing the words that appeared in the title such as life satisfaction.
· In the introduction, we cannot start with an objective, better to write first about the problem and the background of this problem then at the end of the introduction you can write the aim of your study.
· For references no 3, and 4, in academic writing better to summarize their outcome that is more related to your research. I found it unprofessional to just mention the reference and request the reader to see the reference.
· Also reference no 6, “see” should be removed from the whole of the manuscript.
· The style of reference no 7 and 8 is not correct. If the reference comes in the middle of the text it is better to mention the name of the author.
· The introduction overall needs to be rewritten considering writing about the problem and importation of the study compared with previous studies and at the end of the introduction in the last paragraph authors can write their study aim and the significance of this study. In addition, make your research gap clear based on the literature review. With all of this, it is important to correct the reference styles for all research, in this way, you are writing as a report, not a scientific paper.
· In the method section, there is a difference between sampling and data collection producer, so I have many questions:
1- How much is your population?
2- How did you identify your sampling size?
3- Where did you collect your data, and why did you choose this area?
4- What is the sampling technique that you use?
5- You have mentioned the dependent variables in section 2.2 and the independent variable does not exist. I would suggest drawing a model of your study to make it clear.
6- Did you do a pilot study?
7- The authors should add the questionnaire survey to the methodology section.
· In the results, you have long tables with a little bit of explanation. I would suggest dividing them into short tables to explain them independently.
· In the discussion, the comparison with previous studies is missing.
· In conclusion, theoretical and practical contributions should be added.
· Also, the limitations of the study and the suggestions for future researchers were missing.
· Overall, the language and reference styles should be checked.
Author Response
We wish to thank Reviewer 2 for his or her very helpful comments.
Please find our detailed response in the attached file.
Thank you,
Hyun-Chool Lee, Alexandre Repkine and Luwen Zhang

Reviewer 3 Report
Comments and Suggestions for Authors
In summary, the analysis is equitable and effectively encapsulates the importance of issues faced by the elderly population in Korea.
I don't have any objections to the analytic approaches employed in this study
Comments on the Quality of English Language
“The only minor issue I’ve identified is that the author relies excessively on the terms ‘importance’ or ‘important,’ which can introduce subjectivity into the discussion of the issues. Additionally, the use of the term ‘menial’ to describe certain types of work may be uncomfortable for certain readers. I recommend revising the argument to incorporate more objective language and respectful terminology.”
Author Response
We wish to thank Reviewer 3 for his or her very helpful comments.
Please find our detailed response in the attached file.
Thank you,
Hyun-Chool Lee, Alexandre Repkine and Luwen Zhang

Reviewer 4 Report
Comments and Suggestions for Authors
Studying the determinants of health status and life satisfaction among elderly South Koreans is relevant because South Korea is undergoing a remarkable demographic transition, with its population rapidly aging. As the ratio of older citizens continues to rise, there is a pressing need to address issues related to their health and well-being.
General notes:
Throughout the text it may make some sense to replace Koreans by South Koreans and Korea by South Korea.
There are references such as [1], [2], [3], [4], [6], [7], [8], [9], [10], [11]… and others throughout the text where it might make sense to specify the names of the authors.
Original research manuscript.
Studying the determinants of health status and life satisfaction among elderly South Koreans is relevant because South Korea is undergoing a remarkable demographic transition, with its population rapidly aging. As the ratio of older citizens continues to rise, there is a pressing need to address issues related to their health and well-being.
In response to these challenges, the relevance of a health assessment that includes both objective indicators and subjective measures of well-being is increasingly recognised.
Equally important is the assessment of subjective indicators such as life satisfaction and happiness.
The paper falls within the scope of the journal and there are at least twelve references cited of the past five years.
Although some references may be older than the last five years, they provide crucial complements for analysing the issues in question. The main contributions of this document lies in its ability to offer perspectives that can improve the quality of life of the elderly in South Korean society. Its strength lies in the fact that it addresses a contemporary and pertinent issue in the context of ongoing research.
General notes:
Throughout the text it may make some sense to replace Koreans by South Koreans and Korea by South Korea.
There are references such as [1], [2], [3], [4], [6], [7], [8], [9], [10], [11]… and others throughout the text where it might make sense to specify the names of the authors…
Comments on the overall concept:
Article
The Tobit multivariate regression analysis conducted and the interpretations of the various statistical outcomes appear to be suitable.
review
The authors present the problem and the aims of the study in a well contextualised way, looking for evidence that can be extrapolated to the population under study.
The authors present the topic well, based on various studies (they should mention the authors' names).
Specific comments
Line 2-3 - Title: Sugestion: As there are two Koreas, I suggest that the title should describe the target population as older South Koreans; As well as along the text…
Line 19: “…South Koreans”.
Line 21: Keywords: republic of south korea instead of korea; health status indicators instead of health indicators? (https://meshb.nlm.nih.gov/record/ui?ui=D056910 ; https://meshb.nlm.nih.gov/record/ui?ui=D006305). I think Tobit multivariate regression analysis can be removed of keywords.
Introduction:
Line 24: …Older South Koreans
Line 27: South Korea
Lines 29-31: The source of this information should be included and the sentence should be better organised (the meaning is clear but it could be improved).
Lines 32-38: References [1], [2], [3] [4] it might make sense to specify the names of the authors.
Line 43: South Korea
Line 51: South Korean population
…
Line 90: scientific evidence?...
Line 106: source of the survey used?
Lines 114 to 119: Are important issues but for the discussion chapter (is repeated in the discussion)
Note on Introduction: The introduction should define what is meant by ‘older South Koreans’, which corresponds to the population under study.
The final part of the introduction should be clear about the main question(s) of the study.
Line 127 – 161: Materials and Methods: Ethical aspects and how they were safeguarded should be mentioned.
Note on Materials and methods: The variables and their measurements are well explained, justified with references to scientific literature.
Tobit regression is a valuable tool for analysing data with limited or censored observations, providing insights into relationships that may be missed using traditional regression techniques.
The details given are sufficient to replicate the proposed experimental procedures and analysis.
Note on the results: The results appear to be interpreted in an appropriate and justified manner. The sample is well characterised. Initially, a descriptive analysis is presented, characterising the sample with measures of central tendency, as well as the number of observations.
The following is an analysis using Tobi's multivariate regression analysis with very interesting results.
Note on the discussion: The main results and their implications are highlighted, but they can be compared with the results of other studies.
Line 355 – 357: Source of these facts?
Note on the conclusion: The conclusion at the beginning repeats some data from the discussion, so it should be improved. what were the limitations of the study? how do the authors suggest further study of the subject in the light of the results obtained? For exemple how Regular health screenings and check-ups are importante? how to operationalise it in the context of south korea's health system?
Line 432: References with doi do not need to be preceded by https:// and should be in link format to facilitate consultation
Comments on the Quality of English LanguageMinor editing of English language required.
Author Response
We wish to thank Reviewer 4 for his or her very helpful comments.
Please find our detailed response in the attached file.
Thank you,
Hyun-Chool Lee, Alexandre Repkine and Luwen Zhang

Round 2
Reviewer 2 Report
Comments and Suggestions for Authors
I have seen that the authors have taken my comments seriously. However, other minor comments should be considered:
- Discussion of your survey in the introduction paragraph 8 is not appropriate to be here, it is enough to write about it in the method section only.
- In the method section, you mentioned that you followed the random sampling, and then you wrote about “stratification of the sample was based on the Korean Population and Housing Census conducted by the Korean National Statistical Office (NSO)”. So which one is correct? Be accurate.
- Did you do a pilot study by yourself? if not, please justify.
- I know you are following the IEEE reference style but writing the reference in the middle of the text should be like this: According to Rich at al, [10], So the name of the authors should be mentioned before the reference number.
Author Response
We wish to thank Reviewer 2 for his or her useful remarks.
Please see the attached file for our detailed response.
Hyun-Chool Lee and Alexandre Repkine
